# The Feeder Effects of Cultured Rice Cells on the Early Development of Rice Zygotes

**DOI:** 10.3390/ijms242216541

**Published:** 2023-11-20

**Authors:** Yoriko Watanabe, Yuko Nobe, Masato Taoka, Takashi Okamoto

**Affiliations:** 1Department of Biological Sciences, Tokyo Metropolitan University, Minami-Osawa 1-1, Hachioji 192-0397, Tokyo, Japan; delphiniumajacis@icloud.com; 2Department of Chemistry, Tokyo Metropolitan University, Minami-Osawa 1-1, Hachioji 192-0397, Tokyo, Japan; yuuko@tmu.ac.jp (Y.N.); mango@tmu.ac.jp (M.T.)

**Keywords:** amylase, cell wall, feeder cell, fertilization, hydrolytic enzyme, rice, zygote, zygotic development, 2,4-D

## Abstract

Feeder cells and the synthetic auxin 2,4-dichlorophenoxyacetic acid (2,4-D) in a culture medium promote mitosis and cell division in cultured cells. These are also added to nutrient medium for the cultivation of highly active in mitosis and dividing zygotes, produced in vitro or isolated from pollinated ovaries. In the study, an in vitro fertilization (IVF) system was used to study the precise effects of feeder cells and 2,4-D on the growth and development of rice (*Oryza sativa* L.) zygote. The elimination of 2,4-D from the culture medium did not affect the early developmental profiles of the zygotes, but decreased the division rates of multicellular embryos. The omission of feeder cells resulted in defective karyogamy, fusion between male and female nuclei, and the subsequent first division of the cultured zygotes. The culture of zygotes in a conditioned medium corrected developmental disorders. Proteome analyses of the conditioned medium revealed the presence of abundant hydrolases possibly released from the feeder cells. Exogenously applied α-amylase ameliorated karyogamy and promoted zygote development. It is suggested that hydrolytic enzymes, including α-amylase, released from feeder cells may be involved in the progression of zygotic development.

## 1. Introduction

In angiosperms, a zygote, a product of the fertilization of an egg cell with a sperm cell, divides into an asymmetric two-celled embryo [1,2,3,4,5]. In rice, the two-celled embryo continues to divide and develop into a multicellular and then a globular and mature embryo [6,7]. As fertilization and subsequent zygotic development occur in the embryo sac deeply embedded in the ovule of plants, an in vitro fertilization (IVF) system, using microtechniques for three stages, gamete isolation from flowers, the electric fusion of gametes, and a culture of the produced zygote, was applied for a direct and precise analysis of zygotic development [8,9]. Such an IVF system has been established in rice, maize, and wheat [7,10,11,12,13]. A synthetic auxin 2,4-dichlorophenoxyacetic acid (2,4-D) and feeder cells, crucial for the developing IVF-produced zygotes in rice, wheat, and maize, are commonly incorporated in the culture media for the cultivation of somatic cells and protoplasts [14,15,16]. It has been reported that auxins exert diverse effects on cultured somatic cells, such as mitotic activity enhancement, cell elongation/growth, and differentiation [17,18]. The mitotic activities of isolated cells/protoplasts depend/related to cell density in culture [15,16,19,20], and feeder cells are often co-cultured with isolated cells to increase cell density as conditioning factors for growth and development [16,21].

Based on intensive studies on somatic cells/protoplasts, 2,4-D and protein/peptide components present in cell cultures are known to trigger the division and development of somatic cells or protoplasts in vitro, although their division efficiency is low, typically 0.1–0.6% [20,22]. In contrast to somatic cells/protoplasts, zygotes possess totipotent potentials and are highly active with a high division efficiency. It has been indicated that more than 80% of rice and wheat zygotes produced by IVF divide and develop into globular embryos [10,13]. Auxin 2,4-D and feeder cells are supposed to support the developmental potential of cultured zygotes. However, the precise effects of feeder cells and 2,4-D on zygote cultures remain unclear. In this study, we examined the effects of 2,4-D and feeder cells on the developmental profiles of rice zygotes produced by the IVF system and identified α-amylase, possibly secreted by feeder cells, as a candidate conditioning factor in the initial development of rice zygotes.

## 2. Results

### 2.1. 2,4-Dichlorophenoxyacetic Acid Enhances the Division of Embryonic Cells during or after the Multicellular Stage

In this study, a sperm cell isolated from transformed rice plants expressing histone H2B-green fluorescent protein (H2B-GFP) was fused with a wild-type egg cell to monitor the completion of karyogamy, the fusion between male and female nuclei [23], and the division profiles of zygotes [24]. Then, 128 zygotes were produced using IVF and cultured in an N6Z medium containing 2,4-D. Among these 128 zygotes, karyogamy was detected in 105 (Figure 1(Aa,Ab), Table 1) and 98 of the 105 zygotes divided and developed into two-celled embryos (Figure 1(Ac,Ad), Table 1). Subsequently, the two-celled embryos developed mainly into multicellular embryos (Figure 1(Ae,Af)), globular-like embryos (Figure 1(Ag,Ah)), and cell masses (Figure 1(Ai–An), Table 1). Rice zygotes and their subsequent embryos divided and developed with high efficiencies when cultured in the N6Z medium (Table 1), consistent with previous reports [10,25].

Karyogamy was detected in 53 of the 73 zygotes cultured in 2,4-D free N6Z medium (Figure 1(Ba,Bb), Table 1). In total, 47 and 43 of these 53 zygotes divided and developed into two-celled and multicellular embryos, respectively (Figure 1(Bc–Bf), Table 1). Although 28 of the 43 multicellular embryos developed into cell masses (Figure 1(Bg–Bn), Table 1), the remaining 15 showed developmental arrest at the multicellular stage (Figure 1C). These results suggest that 2,4-D functions in the progression of embryonic cell division during or after the multicellular stage. 

### 2.2. Feeder Cells Promote Karyogamy and Subsequent Cell Division in Zygotes

The developmental profiles of zygotes cultured in the feeder cell-free medium were examined to investigate the effects of feeder cells on zygotic development. Among the 103 fertilized eggs cultured in the feeder cell-free N6Z medium, H2B-GFP-labeled zygotic nuclei, an indication of the completion of karyogamy, were detected only in 19 zygotes (Figure 2(Aa,Ab), Table 1). The remaining 84 zygotes appeared to degenerate without karyogamy progression (Figure 2(Ba,Bb)). Of the 19 zygotes with H2B-GFP-labeled nuclei, 5 showed nuclear division but degenerated later (Figure 2(Ca,Cb), Table 1), strongly suggesting that the efficiencies of karyogamy progression and the subsequent first division in zygotes were markedly reduced by the omission of feeder cells from the culture medium. We further tested the culture of zygotes in 2,4-D free N6Z-medium without feeder cells. The developmental profiles of the zygotes were equivalent to those of zygotes cultured in the N6Z medium without feeder cells (Figure 2(Da–Fb), Table 1). Together, these results suggest that the efficiencies of karyogamy progression and zygotic division were reduced in response to feeder cell omission regardless of the presence of 2,4-D. Moreover, karyogamy and subsequent zygotic division are likely promoted by potential substances believed to be secreted by feeder cells.

### 2.3. Heat-Unstable Proteins in Culture Medium Function as Development-Promoting Substances for Zygotes

A conditioned medium was prepared as described in the Section 4, and IVF-produced zygotes were cultured in the medium to examine whether substances secreted from the feeder cells showed positive feeder effects on the progression of zygotic development. Karyogamy was detected in 87 of 104 zygotes (Figure 3(Aa–Ac), Table 1). In total, 72 and 51 of these 87 zygotes were divided into two-celled and multicellular embryos, respectively (Figure 3(Ad–Ai), Table 1). In the conditioned medium, the frequency of zygotes that completed karyogamy and developed into two-celled/multicellular embryos was greatly increased compared to that of zygotes cultured without feeder cells (Table 1). Although 8 of the 51 multicellular embryos progressed into the cell mass stage (Figure 3(Aj–Ar)), the remaining embryos showed arrested growth (Figure 3B, Table 1). These results suggest that the conditioned medium rescued the efficiencies of karyogamy and subsequent zygotic development and that the conditioned medium contained potential development-promoting substances (DPSs) secreted by the cultured Oc cells.

Next, to determine the molecular nature of DPSs, we tested their heat stability by culturing the zygotes in a preheated or non-heated control-conditioned medium. In the case of zygote culture in the non-heated conditioned medium, karyogamy was detected in 24 of the 39 zygotes tested (Figure 4(Aa–Ac), Table 2). In addition, 11 of these 24 zygotes divided and developed into two-celled embryos, with 3 multicellular embryos being formed (Figure 4(Ba–Bc,Ca–Cc), Table 2). Karyogamy was detected in 16 of the 37 zygotes cultured in the heat-treated conditioned medium (Figure 4(Da–Dc), Table 2). However, none of the 16 zygotes showed a division profile during culture and degenerated without division (Table 2). These results suggest that heat treatment of the conditioned medium decreased the activities of potential DPSs and that potential DPSs are heat-unstable protein components.

### 2.4. Hydrolytic Enzymes Were Identified as Potential DPSs for Zygotes

Protein fractions prepared from heat-treated and non-heat-conditioned medium were separated using sodium dodecyl sulfate-polyacrylamide gel electrophoresis to identify heat-unstable proteins in the conditioned medium. The proteins were subjected to in-gel digestion, and the subsequent peptides were analyzed using liquid chromatography coupled with tandem mass spectroscopy. A total of 978 proteins were identified using duplicate proteomic analyses. Among these proteins, ubiquitin proteins were frequently detected in both heat-treated and non-heat-treated media at comparable levels, consistent with the heat-stable nature of ubiquitin (Table 3) [26,27]. Moreover, it provided a quantitative control for a comparative analysis of the protein abundance between the two kinds of conditioned media. As candidate DPSs, 21 proteins detected exclusively or highly preferentially in the non-heat-conditioned medium with a total peptide count of >20 were selected (Table 4). Of these proteins, 15 were hydrolytic enzymes possessing a putative signal peptide for secretion, and 12 of these 15 hydrolytic enzymes belonged to the glycosidase family. Among these glycosidases, α-amylase was most abundantly detected (Table 4); therefore, we attempted to examine the effects of α-amylase in promoting zygotic development.

### 2.5. Exogenously Applied α-Amylase Partially Promoted the Initial Development of Zygotes

IVF-produced zygotes were cultured in feeder cell and 2,4-D free N6Z medium to which α-amylase derived from *Bacillus licheniformis* was exogenously added at the concentrations of 017, 1.7, or 17 U/mL to evaluate the effects of α-amylase on zygotic development. For the α-amylase treatments at the 0.17 and 17 U/mL concentrations, the developmental efficiencies in karyogamy progression were comparable to that of the control (Figure 5(Aa–Ac,Ba–Bc,Ea–Ec,Fa–Fc), Table 5). However, efficiency in karyogamy progression increased approximately 2 folds when α-amylase was added at the concentration of 1.7 U/mL (Figure 5(Ca–Cc,Da–Dc), Table 5). These results suggest that α-amylase exogenously applied at appropriate concentrations partially functions as a DPS for zygotic development.

## 3. Discussion

The present study provided the possibility that hydrolases, especially α-amylase, secreted from feeder cells are involved in the initial developmental steps of rice zygotes cultured in vitro. In the somatic embryogenesis of orchard grass, α-amylase is transiently expressed during early embryo development up to the globular stage and localizes on the cell surface and cell-to-cell adhesion zone in microcluster cells, the earliest morphological structures during somatic embryogenesis [28,29]. In addition, the expression of α-amylase in the somatic embryos of orchard grass is speculated to relate to the embryonic potential of cell mass, although the mechanisms underlying the α-amylase-mediated determination of embryonic character remains to be elucidated [28,29,30]. It is likely that the α-amylase detected in our study also degraded the polysaccharides existing in the cell wall or plasma membrane of the rice zygotes. The determination of the target substance of α-amylase and the responsive machineries in rice zygotes against α-amylase will be required for understanding how the enzyme functions as DPS on rice zygotes.

Hydrolases have been reported as possible conditioning factors [21,28,31,32]. In the present study, classIII chitinase RCB4 was also identified as possible hydrolytic enzyme possessing DPS function (Table 4). The chitinase RCB4 has been purified from rice (*O. sativa*, Nakdong) cell suspension and characterized as endochitinase with an optimum pH around 4 [33]. Van Hengel et al. (2001) reported that endochitinase acts on the arabinogalactan proteins (AGPs) of the cell wall [22]. Somatic embryogenesis-incompetent cells regain their potential when chitinase-treated AGPs are added to the culture medium [22,34,35], suggesting that AGPs degraded by endochitinase act as signaling molecules to trigger somatic embryogenesis. Immunological assays using antibodies against AGPs have shown that AGPs are abundantly localized in the cells of the rice embryo sac, such as antipodal cells, egg cells, and synergids [36]. In tobacco, Qin and Zhao (2006) reported abundant AGP localization in egg cells and a subsequent APG decrease in zygotes after fertilization [37], suggesting the involvement of AGPs in embryonic development. These suggest that chitinase RCB4 secreted from feeder cells enhances the development of rice zygotes through the possible degradation of APGs in the cell walls around rice zygotes.

Cell wall proteins are not abundant in quantity but are believed to have crucial functions in cell defense, cell wall structure and modification, interactions with plasma membrane proteins, and signal transduction [28]. Zygotes start forming cell walls rapidly after gamete fusion because female and male gametes lack the cell walls to facilitate gamete fusion [38,39]. In addition, it has been reported the cell wall of tobacco zygotes possesses essential functions for maintaining cell polarity, the apical–basal axis, and typical suspensor formation [40]. Certain hydrolytic enzymes, including α-amylase, may be involved in the proper cell wall formation in plant zygotes and subsequent zygotic development.

## 4. Materials and Methods

### 4.1. Plant Materials

*Oryza sativa* L. cv Nipponbare plants were grown in an environmental chamber (K30-7248; Koito Industries, Yokohama, Japan) at 26 °C under a 13 h light/11 h dark photoperiod. Transformed rice plants expressing the histone H2B-green fluorescent protein (H2B-GFP) fusion protein were prepared [41].

### 4.2. Isolation of Gametes, Electro-Fusion of Gametes, and Subsequent Culture of Zygotes

The isolation of egg and sperm cells from rice flowers and the subsequent electro-fusion of isolated gametes for zygote production were conducted [10,42]. The resulting zygotes were cultured as previously reported [10], with a few modifications. For examining the effects of feeder cells on zygotic development, the zygotes were cultured in the N6Z medium [10,12], with or without feeder cells (rice suspension cell culture, line Oc, provided by Riken Bio-Resource Center, Tsukuba, Japan; https://plant.rtc.riken.jp/resource/cell_line/web_documents/cell_lines/rpc00031.html, 15 November 2023). For analyzing the effects of 2,4-dichlorophenoxyacetic acid (2,4-D) on zygotic development, zygotes were cultured in N6Z medium with or without 2,4-D. For examining the effects of α-amylase on zygotic development, zygotes were cultured in feeder cell- and 2,4-D-free N6Z medium containing 0.17, 1.7, or 17 U/mL of *Bacillus licheniformis* α-amylase purchased from Sigma, St. Louis, MO, USA.

### 4.3. Microscopic Observations

Zygotes expressing H2B-GFP were observed under an IX-71 inverted fluorescence microscope (Olympus, Tokyo, Japan) with 460–490 nm excitation and 510–550 nm emission wavelengths (U-MWIBA2 mirror unit; Olympus, Tokyo, Japan). Digital images of the zygotes and their resulting embryos were obtained using a cooled charge-coupled device camera (DP73; Olympus) with cellSens software (Standard version 1.6, Olympus).

### 4.4. Preparation of Conditioned Medium

The Oc cells (feeder cells) in culture were subcultured every week according to the protocol provided by Riken Bio-Resource Center (https://plant.rtc.riken.jp/resource/cell_line/web_documents/cell_lines/rpc00031.html, 15 November 2023). The Oc cells at 4 or 6 days after subculture were weighed (0.2–0.6 g) after rinsing them with 2,4-D-free MS medium and removing the medium using pipettes. Subsequently, the cells were suspended in 10 mL of 2,4-D-free N6Z medium and cultured overnight in 6 cm diameter Petri dishes at 26 °C in darkness with rotation at 40 rpm. The cultured medium was roughly separated from feeder cells and filtered through a filter (0.45 µm pore size, Millex-HV Filter Unit). The filtrate was used as the conditioned medium.

### 4.5. Preparation of Heat-Treated Conditioned Medium

The conditioned medium was dispensed into two glass vials of 5 mL each. One aliquot was placed in boiling water at 100 °C for 15 min and then cooled to room temperature to be used as the heat-treated conditioned medium. Another aliquot of the conditioned medium was left at room temperature while preparing the heat-treated medium and used as the non-heat-treated conditioned medium.

### 4.6. Isolation of Protein Samples from the Conditioned Medium for Proteome Analysis

The conditioned medium with or without heat treatment was centrifuged at 15,000× *g* for 15 min, and 300 µL of methanol was added to 100 µL of the supernatant. After centrifugation at 15,000× *g* for 3 min, the precipitate was dried under vacuum and used as the protein fraction from the conditioned medium. The dried precipitate dissolved in 20 µL sample buffer solution (2-mercaptoethanol+; FUJIFILM Wako Chemicals, Oosaka, Japan) was subjected to 100 °C for 5 min and applied to sodium dodecyl sulfate-polyacrylamide gel electrophoresis (SDS-PAGE).

### 4.7. SDS-PAGE and Silver Staining

According to Laemmli (1970) [43], 10–20% SDS-PAGE gels were prepared in a small mold (e-PAGEL HR her-T1020L, 90 mm × 83 mm × 1 mm; Atto, Tokyo, Japan), and 20 µL protein sample solutions were applied. After electrophoresis, proteins in the gels were detected using conventional silver staining [44]. When used for subsequent liquid chromatography coupled with tandem mass spectroscopy (LC-MS/MS) analyses, the proteins in the gel were visualized with modified silver staining using Silver Stain MS Kit (FUJIFILM Wako Chemicals).

### 4.8. Identification of Proteins by Tandem Mass Spectrometry

SDS-PAGE gels were cut into five pieces. Proteins in each piece were in-gel-digested with trypsin [45] and identified with LC-MS/MS analyses using a direct nanoflow LC-MS system equipped with an Orbitrap mass spectrometer (Q Exactive; Thermo Scientific, Waltham, MA, USA), as described elsewhere [46]. The dataset of protein sequences obtained from the Rice Annotation Project Database (Tsukuba, Japan; http://rapdb.dna.affrc.go.jp/download/irgsp1.html, 19 November 2019) was searched using Mascot software (ver. 2.3.2; Matrix Science, Boston, MA, USA) with the following parameters. The variable modification parameters were pyro-Glu, acetylation (protein N-terminus), and oxidation (Met). The maximum missed cleavage was set at 3 with a peptide mass tolerance of ±15 ppm. Peptide charges from +2 to +4 states and MS/MS tolerances of ±0.8 Da were allowed. The criteria for peptide identification were based on the vendor’s definitions (expectation value < 0.05; Matrix Science), and we assigned the protein as “identified” when at least two peptides were identified from the protein.

## Figures and Tables

**Figure 1 ijms-24-16541-f001:**
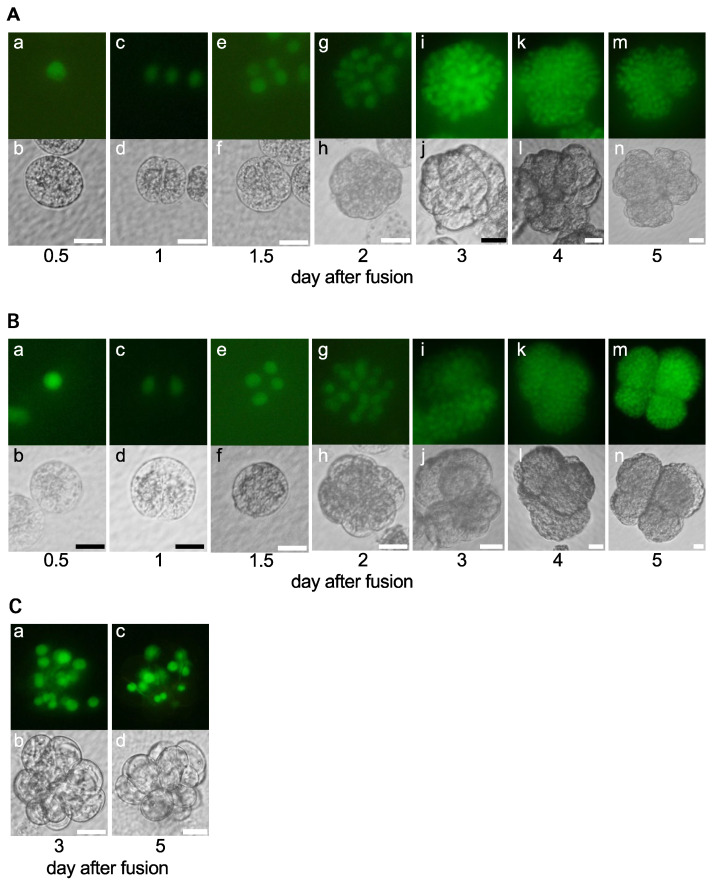
Developmental profiles of zygotes cultured in N6Z medium (**A**) and 2,4-D free N6Z medium (**B**,**C**). (**A**) Developmental profiles of the zygotes cultured in N6Z medium. After completion of karyogamy in zygote (**Aa**,**Ab**), the zygote developed into a two-celled embryo (**Ac**,**Ad**), multicellular embryo (**Ae**,**Af**), a globular-like embryo (**Ag**,**Ah**), and cell mass (**Ai**–**An**). (**B**) Developmental profiles of the zygotes cultured in 2,4-D free N6Z medium. After completion of karyogamy in zygote (**Ba**,**Bb**), the zygote developed into a two-celled embryo (**Bc**,**Bd**), multicellular embryo (**Be**,**Bf**), a globular-like embryo (**Bg**,**Bh**), and cell mass (**Bi**–**Bn**). (**C**) Developmentally arrested embryos cultured in 2,4-D free N6Z medium. Upper and lower panels are fluorescent and bright-field images, respectively. Bars, 20 µm.

**Figure 2 ijms-24-16541-f002:**
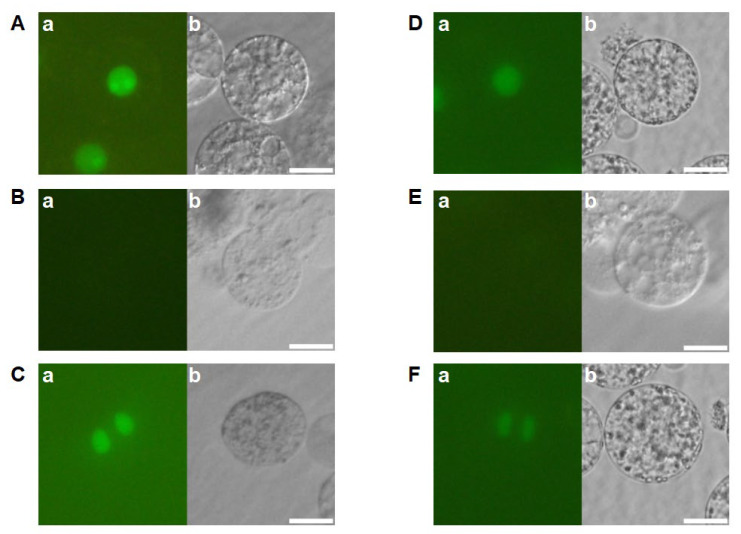
Developmental profiles of zygotes cultured in feeder cell free N6Z medium. Zygotes cultured in feeder-cell free N6Z medium (**A**–**C**) and in feeder cell and 2,4-D free N6Z medium (**D**–**F**) were observed 1 day after fusion. Left and right panels are fluorescent and bright-field images, respectively. Bars, 20 µm.

**Figure 3 ijms-24-16541-f003:**
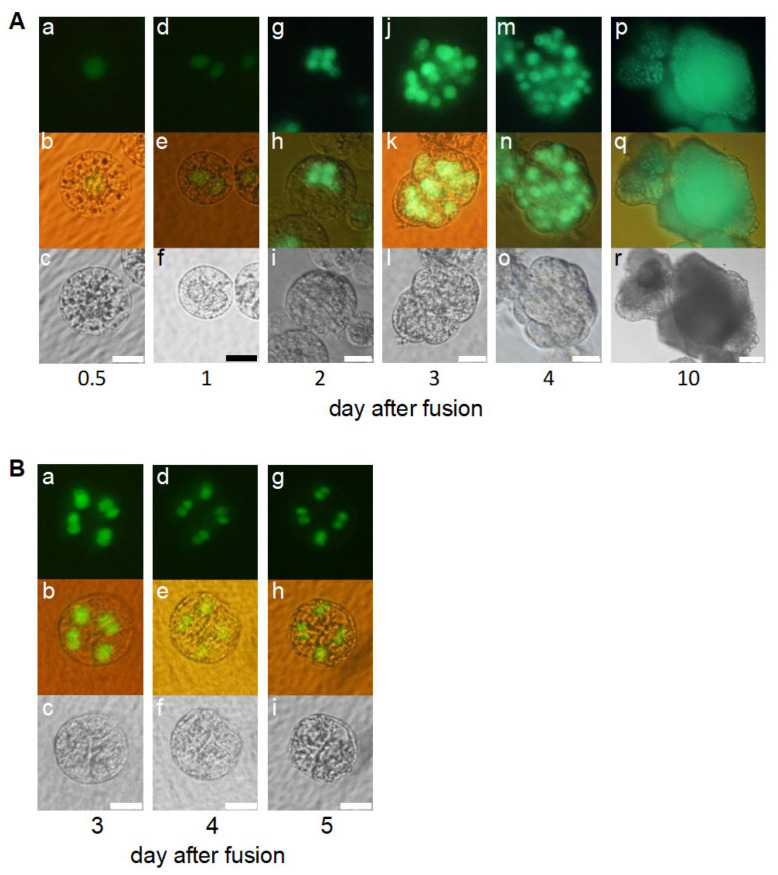
Developmental profiles of the zygotes cultured in conditioned medium. (**A**) After completion of karyogamy in zygote (**Aa**–**Ac**), the zygote developed into a two-celled embryo (**Ad**–**Af**), multicellular embryo (**Ag**–**Ai**), a globular-like embryo (**Aj**–**Ao**), and cell mass (**Ap**–**Ar**). Bars, 20 µm (**Aa**–**Ao**), 100 µm (**Ap**–**Ar**). (**B**) The zygote arrested development. The zygote developed into a multicellular embryo (**Ba**–**Bc**) 3 days after fusion and arrested development thereafter (**Bd**–**Bi**). Upper, lower, and middle panels are fluorescent, bright-field, and merged images, respectively. Bars, 20 µm.

**Figure 4 ijms-24-16541-f004:**
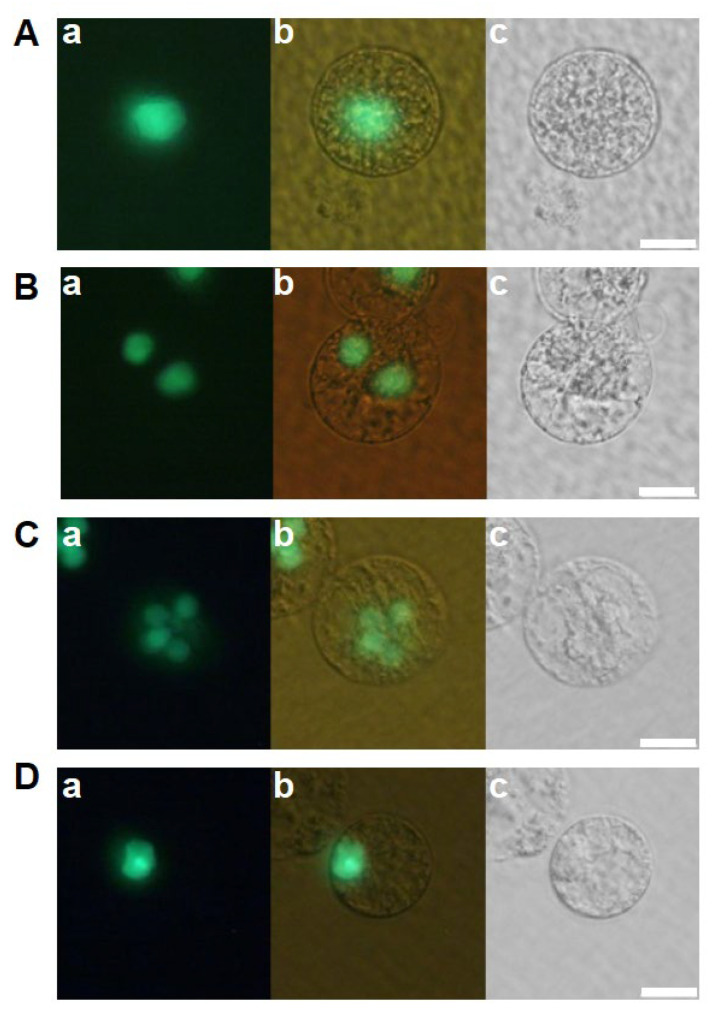
Developmental profiles of the zygotes cultured in conditioned medium without heat treatment (**A**–**C**) and in heat-treated conditioned medium (**D**). After completion of karyogamy in zygote cultured in non-treated conditioned medium (**Aa**–**Ac**), the zygote developed into a two-celled embryos (**Bc**–**Bc**) and multicellular embryos (**Ca**–**Cc**). In case of zygotes cultured in heat-treated conditioned medium, zygotes were degenerated after completion of karyogamy (**Da**–**Dc**). Left, right, and middle panels are fluorescent, bright-field, and merged images, respectively. Bars, 20 µm.

**Figure 5 ijms-24-16541-f005:**
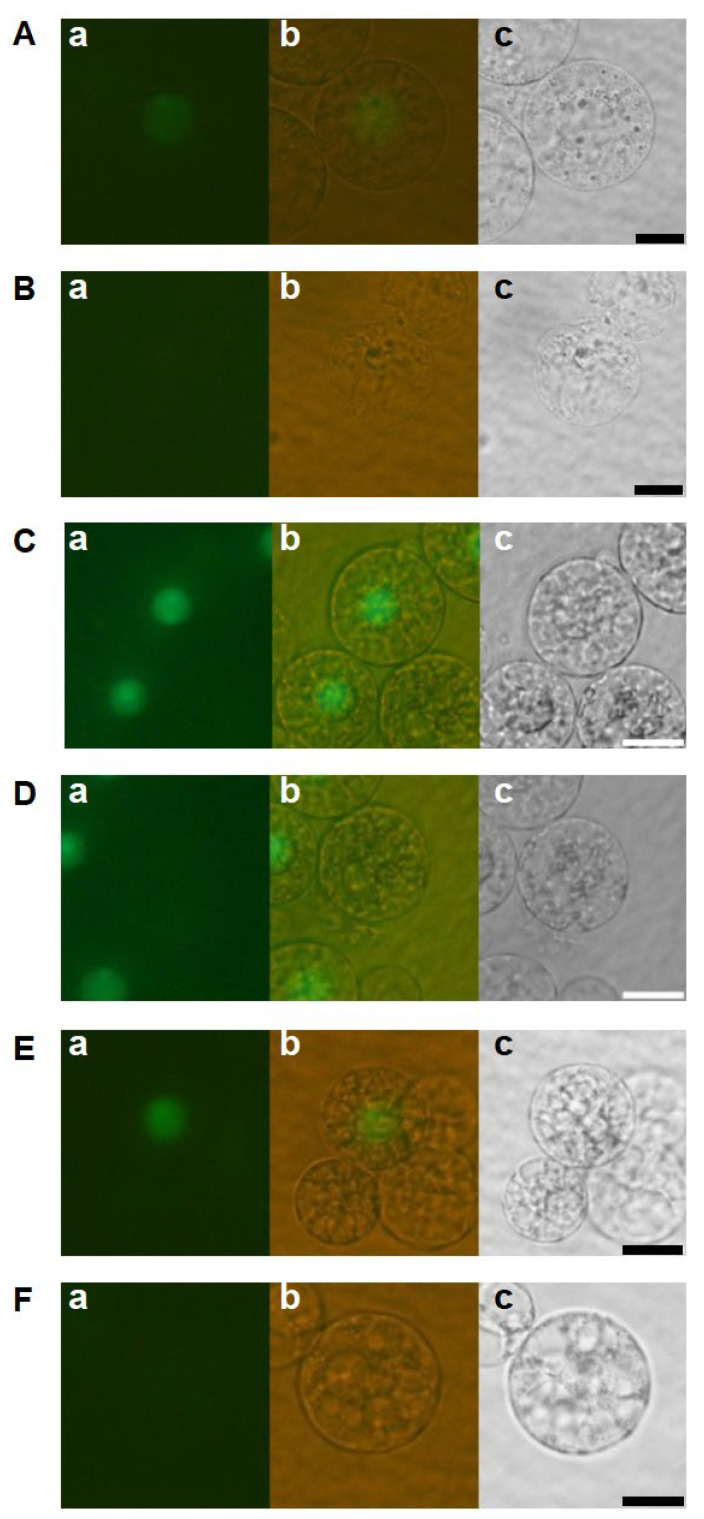
Effect of exogeneously applied α-amylase on the development of rice zygotes. Zygotes were cultured in 2,4-D free N6Z medium containing α-amylase at the concentration of 0.17 U/mL (**A**,**B**), 1.7 U/mL (**C**,**D**), and 17 U/mL (**E**,**F**). Left, right, and middle panels are fluorescent, bright-field, and merged images, respectively. Bars, 20 µm.

**Table 1 ijms-24-16541-t001:** Effects of 2,4-D and feeder cells on developmental profiles of rice zygotes.

Medium		No. of Zygotes Cultured	No. of Zygotes That Developed to Specific Growth Stages				
Feeder Cell	2,4-D		Karyogamy	Two-Celled Embryo	Multicellular Embryo	Globular-LikeEmbryo	Cell Mass
+	+	128	105	98	97	88	74
+	−	73	53	47	43	31	28
−	+	103	19	5	0	0	0
−	−	83	27	5	0	0	0
Conditioned medium		104	8	72	51	12	8

**Table 2 ijms-24-16541-t002:** Developmental profiles of zygotes cultured in conditioned medium pretreated with or without heat.

Heat Treatment ^a^	No. of Zygotes Cultured	No. of Zygotes That Developed to Specific Growth Stages		
		Karyogamy	Two-Celled Embryo	Multicellular Embryo
−	39	24	11	3
+	37	16	0	0

^a^ Conditioned medium with or without heat-treatment at 100 °C for 15 min were used for zygote culture.

**Table 3 ijms-24-16541-t003:** Proteins detected both in non-heat and heated conditioned medium with >20 spectra.

Gene Locus	Protein ^a^	Number of Identified Spectra ^b^	
		Non-Heat Conditioned Medium	Heated Conditioned Medium
Os06g0681400	Ubiquitin domain-containing protein.	368	310
Os02g0161900	Similar to polyubiquitin containing seven ubiquitin monomers.	317	268
Os04g0628100	Similar to polyubiquitin.	250	207
Os06g0673500	Similar to polyubiquitin containing seven ubiquitin monomers.	93	79
Os03g0770775	Hypothetical protein.	61	78
Os01g0328400	Ubiquitin.	62	58
Os05g0160200	Ubiquitin.	62	58
Os06g0650100	Similar to the ubiquitin-NEDD8-like protein RUB2.	62	58
Os07g0489500	Ubiquitin domain-containing protein.	62	58
Os09g0420800	Similar to ubiquitin.	62	58
Os09g0452700	Ubiquitin.	62	58
Os01g0687400	Similar to chitinase.	157	47
Os03g0429000	Proteinase inhibitor I25, cystatin domain-containing protein.	45	41
Os10g0359200	Similar to UPF0496 protein 1.	30	36
Os01g0853000	Conserved hypothetical protein.	29	35
Os02g0525900	Similar to acetyl-coenzyme A synthetase 2 (acetate–CoA ligase 2) (acyl-activating enzyme 2).	43	29
Os11g0427700	Xanthine/uracil/vitamin C permease family protein.	25	29
Os01g0582400	Similar to multidomain cyclophilin type peptidyl-prolyl cis-trans isomerase.	28	26
Os08g0434100	Similar to S-like ribonuclease (RNase PD2) (fragment).	34	22
Os06g0208800	Lysin motif-containing protein, pattern recognition receptor, roles in peptidoglycan and chitin perception in innate immunity	36	21
Os03g0786100	Similar to glycolate oxidase (fragment).	24	20

^a^ Protein annotations were obtained from The Rice Annotation Project Database (RAP-DB, Tsukuba, Japan http://rapdb.dna.affrc.go.jp/download/irgsp1.html, 19 November 2019). ^b^ The number of spectra is the sum of analyses performed twice. Abbreviations: RUB2, Related to Ubiquitin 2; NEDD8, neural precursor cell-expressed, developmentally downregulated 8.

**Table 4 ijms-24-16541-t004:** Proteins that were preferentially detected from the non-heat-conditioned medium with >20 spectra.

Gene Locus	Protein ^a^	Number of Identified Spectra ^b^		Signal Peptide ^c^
		Non-Heat-Conditioned Medium	Heated Conditioned Medium	
Os08g0473900	Similar to alpha amylase isozyme 3D.	531	2	+
Os08g0473600	Alpha-amylase isozyme 3E precursor.	259	0	+
Os09g0457800	Alpha-amylase isozyme 3C precursor.	135	0	+
Os09g0457600	Alpha-amylase isozyme 3B precursor.	131	0	+
Os04g0574200	FAS1 domain-containing protein	130	13	+
Os06g0104300	Similar to pectinesterase-like protein.	107	0	+
Os03g0761500	Similar to subtilisin-like protease (fragment).	73	0	+
Os03g0603600	PLC-like phosphodiesterase, TIM beta/alpha-barrel domain-containing protein.	66	0	+
Os02g0121300	Cyclophilin, peptidyl-prolyl *cis-trans* isomerase, auxin signal transduction, lateral root initiation, stress tolerance	48	0	−
Os11g0525600	Similar to alpha-mannosidase.	46	0	+
Os01g0739700	Glycoside hydrolase, family 17 protein.	43	0	+
Os10g0493600	Alpha-galactosidase precursor.	43	0	+
Os02g0765400	Similar to alpha-amylase.	36	0	+
Os02g0765600	Alpha-amylase glycoprotein, degradation of starch granule	36	0	+
Os06g0546500	Similar to class III peroxidase GvPx2b (fragment).	32	0	+
Os02g0139300	Glycoside hydrolase, family 17 protein.	28	0	+
Os10g0416100	Class III chitinase RCB4.	26	0	+
Os03g0703100	Similar to beta-glucosidase.	25	0	+
Os03g0327600	Ricin B-related lectin domain-containing protein.	24	0	−
Os03g0285700	Similar to L-ascorbate peroxidase.	21	1	−
Os09g0548200	Peptidoglycan-binding lysin subgroup domain-containing protein.	21	2	+

^a^ Protein annotations were obtained from The Rice Annotation Project Database (RAP-DB, Tsukuba, Japan http://rapdb.dna.affrc.go.jp/download/irgsp1.html, 19 November 2019). ^b^ The number of spectra is the sum of analyses performed twice. ^c^ Existence of signal peptides for protein secretion on the identified proteins was estimated using TargetP-2.0 (https://services.healthtech.dtu.dk/services/TargetP-2.0/, 24 July 2023). Abbreviations: FAS1, fasciclin-1; PLC, phospholipase C; TIM, triose-phosphate isomerase.

**Table 5 ijms-24-16541-t005:** Effects of exogenously applied α-amylase on developmental profiles of rice zygotes.

α-Amylase Concentration (Unit/mL)	No. of Zygotes Produced	No. of Zygotes That Developed to Specific Growth Stages	
		Karyogamy	Two-Celled Embryo
0	13	3	0
0.17	7	2	0
1.7	18	12	0
17	21	5	0

## Data Availability

The datasets generated and/or analyzed during the current study are available from the corresponding author on reasonable request.

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
