# Peer review of "The Feeder Effects of Cultured Rice Cells on the Early Development of Rice Zygotes"

_ijms, 2023, doi:10.3390/ijms242216541_

Round 1
Reviewer 1 Report
Comments and Suggestions for Authors
I do not recommend this manuscript of paper in the present form. The abstract/introduction and the discussions lack focus on the subject of research done and described in this manuscript. The entire manuscript may be revised and restructured. I have attempted restructuring of the abstract and the introduction. The distinction between introduction and discussion are missing. The discussion part of this manuscript need special attention. This must be thoroughly revised. The experiments designed and results obtained are important. The illustration are impressive.

Reviewer 2 Report
Comments and Suggestions for Authors
Dear Authors,
This manuscript addresses the effect of the feeder layer of rice cells on the development of the rice zygotes in combination with synthetic auxin, and it is an excellent continuation of published work from Dr. Okamoto from 2007 and 2006.
Manuscript title:
Feeder effects of cultured rice cells on the early development of rice zygotes.
Abstract: Can be improved, and part of it is more suitable for the Discussion section.
Please explain in detail the term karyogamy.
Introduction: In this section, there is a lot of old literature cited, and, e.g., new work done with different crop and tree species, especially with alfa amylases, relevant also for this work is missing. Please add it here.
Results: Well structured and presented.
Line 116: Table 1, please explain what is Oc.
Line 193: Table 2, please explain what kind of heat treatment has been used. No heat treatment is mentioned as the goal of this study or in the introduction section.
Line 356: Bacillus licheniformis needs to be written in Italic script! Please correct it!
All Tables are missing captions!
Discussion: is very short, and can be improved. Also
Some of the references are present in the section results, so it is making reading and orientation for the reader a little bit confusing!
Also, except for only citation 39 (20219) in this section, the other references are quite old. There are new publications available, especially about the role of alfa-amylase in the process of somatic embryogenesis. Please add here a new publication.
None of the results from the experiment with proteins obtained from the non-heat and heated conditioned media are adequately discussed in this section. Please include this part in the discussion properly
M&M:
Line 449: specify Oc line.
Line 454: Latin name needs Italic script!
Line 464: section 4.4 Preparation of the condition medium
Please add information on how often these cells were subcultured and how old cells were used for the feeder.
In what medium and culture conditions are these suspension cells growing?
Did you also test different rice genotypes for the feeder layers?
Line 479: 4.5 Section preparation of heat –tread condition medium
Please clearly specify the temperature used in this part of the experiments! As it is written now, no one can repeat your experiment !!
This is an exciting manuscript describing an interesting topic relevant for many scientists working not only with rice but also needs real improvement before publishing.
7.11.2023
Round 2
Reviewer 1 Report
Comments and Suggestions for Authors
The authors have incorporated suggestions and revised the manuscript satisfactorily .
Reviewer 2 Report
Comments and Suggestions for Authors
Thank you for reviewing and correcting the manuscripts.
I think now the ms can be accepted for publishing.
14.11.2023